# Factors Related to Physician Burnout and Its Consequences: A Review

**DOI:** 10.3390/bs8110098

**Published:** 2018-10-25

**Authors:** Rikinkumar S. Patel, Ramya Bachu, Archana Adikey, Meryem Malik, Mansi Shah

**Affiliations:** 1Department of Psychiatry, Griffin Memorial Hospital, 900 E Main St, Norman, OK 73071, USA; 2Department of Medicine, Providence Hospital, Washington, DC 20017, USA; ramyabachu1@gmail.com; 3Department of Psychiatry, Zucker Hillside Hospital, Glens Oaks, NY 11004, USA; adikey.archana@gmail.com (A.A.); shahmansi717@gmail.com (M.S.); 4Department of Psychiatry, Mindcare Clinic, Lincolnwood, IL 60712, USA; meryem.malik@gmail.com

**Keywords:** burnout, consequences, causes, depression, patient care, healthcare impact

## Abstract

Physician burnout is a universal dilemma that is seen in healthcare professionals, particularly physicians, and is characterized by emotional exhaustion, depersonalization, and a feeling of low personal accomplishment. In this review, we discuss the contributing factors leading to physician burnout and its consequences for the physician’s health, patient outcomes, and the healthcare system. Physicians face daily challenges in providing care to their patients, and burnout may be from increased stress levels in overworked physicians. Additionally, the healthcare system mandates physicians to keep a meticulous record of their physician-patient encounters along with clerical responsibilities. Physicians are not well-trained in managing clerical duties, and this might shift their focus from solely caring for their patients. This can be addressed by the systematic application of evidence-based interventions, including but not limited to group interventions, mindfulness training, assertiveness training, facilitated discussion groups, and promoting a healthy work environment.

## 1. Introduction

Burnout is a psychological syndrome characterized by emotional exhaustion, depersonalization, and a sense of reduced accomplishment in day-to-day work [1]. Among physicians, emotional exhaustion refers to feelings of being overextended and the depletion of one’s emotional and physical resources, making them feel drained and “used up”. This can sometimes lead to negative, cynical, hostile attitudes and detached feelings towards patients, known as depersonalization, and treating their patients as objects rather than human beings. Reduced personal accomplishment implies negative self-appraisal, feelings of incompetence, and inefficiency in daily work. Numerous studies have shown that 25%–60% of physicians report exhaustion across various specialties [2,3,4]. Changes in the healthcare system have created marked and growing external pressures among physicians. Furthermore, physicians are predisposed to burnout due to inherent traits such as compulsiveness, guilt, self-denial, and working in a medical culture that emphasizes perfectionism, denial of personal vulnerability, and delayed gratification [2].

A nationwide survey study conducted by Shanafelt et al. evaluated the prevalence of stress in physicians and included 6880 US physicians (aged 35–60 years) from different specialties. The results of the survey showed that 54.4% of the US physicians reported at least one symptom of burnout when compared with 45.5% in 2011, and satisfaction with work–life balance in physicians had declined between 2011 (48.5%) and 2014 (40.9%) [5]. Physicians working in specialties at the front line of the care access (e.g., emergency medicine, general internal medicine, neurology, and family medicine) are at highest risk of stress. When compared with other professions, physicians have nearly twice the risk of burnout and work–life dissatisfaction after controlling for factors such as work hours and level of education [6]. Physician fatigue has a negative impact not only on one’s well-being but also on patient care and the health care system. This may be consistent with low job satisfaction, decreased work productivity, medical errors, poor quality of patient care, low job satisfaction, early retirement, and healthcare system failure [7,8].

In this article, we provide a detailed discussion of the contributors and consequences of physician burnout and its impact on patient care, physician health, and the health care system after reviewing the studies published in peer-reviewed scientific journals. We also make certain recommendations for hospital managers and the healthcare system to overcome the effects of burnout in the physicians.

## 2. Methods

Literature searches (PubMed, EMBASE, and Google Scholar) were carried out (2000 to 2018) using the keyword “burnout” and cross-referencing it with “physicians”, “depression”, “healthcare system”, “consequences”, “contributors”, and “patient care”. Articles found through this indexed search were reviewed and manually screened to identify relevant studies including longitudinal case–control studies, cross-sectional and cohort studies, and systematic reviews.

## 3. Contributors of Physician Burnout

Burnout usually results from excess work-related stressors. Risk factors associated with it can be divided into categories such as work factors, personal characteristics, and organizational factors. 

### 3.1. Work Factors

In general, work factors that contribute to physician burnout include excessive workloads, long working hours, specialty choice, frequent call duties (night call or weekend call), comprehensive documentation in electronic medical records, time spent at home on work-related factors, risk of malpractice suits, and methods physicians use to deal with patient death and illness [7]. Most studies suggest that physicians find the loss of autonomy at work, decreased control over the work environment, inefficient use of time due to administrative requirements, and loss of support from colleagues to be the central factors. A national survey conducted among physicians across all specialties in 2014 found that physicians who used electronic health records (EHRs) and computerized physician order entry (CPOE) were less satisfied with the time spent on clerical work and were at increased risk of professional exhaustion [9]. For every hour spent on patient interaction, the physician has an added one-to-two hours finishing the progress notes, ordering labs, prescribing medications, and reviewing results without extra compensation [10].

### 3.2. Personal Characteristics

Personal characteristics associated with burnout include being self-critical, engaging in unhelpful coping strategies, sleep deprivation, over commitment, perfectionism, idealism and work–life imbalance, and an inadequate support system outside the work environment (e.g., having no spouse, partner, or children) [11]. Burnout was once thought to be a late-career phenomenon, but the recent studies suggest that younger physicians have nearly twice the risk of stress compared with older colleagues and that onset may be as early as residency training [12]. Although gender is not an independent predictor of burnout, after adjusting for age and other factors some studies have found female physicians to have 20%–60% increased odds of fatigue compared to men [7,9,13]. Females are more likely to experience burnout because of the strong influence of emotional exhaustion on depersonalization, which can further lead to reduced personal accomplishment [14]. A Norwegian study reporting the risk factors for physician burnout found higher exhaustion levels among women resulted from work–home conflicts, whereas for men, fatigue was strongly predicted by work load [15]. Other factors suggest that having a child younger than 21 years old increases the odds of burnout by 54%, and having a partner or spouse in a non-medical profession increases the odds by 23% [7]. It could be certain that personality types like neurotic individuals have a higher risk of burnout while extraverted, conscientious, and agreeable individuals are less likely to demonstrate symptoms of burnout [8]. 

### 3.3. Organizational Factors

Organizational factors such as negative leadership behaviors, work load expectations, insufficient rewards, limited interpersonal collaboration, and limited opportunities for advancement and social support for physicians may also influence burnout [16]. Some studies have shown that organizations and leaders that provide physicians with increased control over the workplace issues, and are more “physician friendly” and “family friendly”, are more likely to employ physicians with higher career satisfaction and lower reported stress [17]. 

## 4. Consequences of Patient Care and Health Care Costs

Most physicians argue that the administrative burden is overriding the quality time required for assessing patients [10]. It is conceivable that the clerical work intrudes on the physician’s time, resulting in emotional exhaustion, depersonalization or a diminished sense of personal success, substance abuse, depression, post-traumatic stress disorder (PTSD), and suicidality [8,10]. Emotional exhaustion forms the core of physician burnout. Physicians working in outpatient settings experienced higher burnout compared to those working in inpatient facilities [18]. The foremost branches at risk are internal medicine, neurology, and emergency medicine, amongst which 45% are critical care physicians experiencing severe burnout syndrome symptoms and 71% are pediatric critical care physicians, followed by emergency medicine physicians [8]. About 9% of the physicians who experience burnout are prone to have made at least one major medical error in the past three months and receive low patient–physician satisfaction score [8,10]. There is a strong bidirectional dose–response relationship between burnout syndrome scores and medical errors, where errors lead to distress and distress leads to errors [8]. Emotional exhaustion is suggestive of being positively correlated with the workload, constrained organization, work conflicts, violence, and poor mental health, and negatively correlated with autonomy. It is seen that female physicians are more prone because of clinical burden. Unable to handle this challenge, some physicians leave the organization, resulting in a loss of $50,000 to $1 million in training and recruiting a new physician [8,10]. US physicians experience lower emotional exhaustion compared to European physicians due to safe culture, excellent career opportunities, and the problem facing coping methods [18].

Tawfik et al. [19] conducted a population-based survey of US physicians from August 2014 to October 2014 with the objective of evaluating physician burnout, well-being, and work unit safety grades about perceived significant medical errors. The results of the survey showed that 54.3% reported symptoms of fatigue, 32.8% reported excessive fatigue, and 6.5% reported recent suicidal ideation, with 3.9% reporting a poor or failing patient safety grade in their work area and 10.5% reporting a significant medical error in the three months prior. Physicians reporting medical errors were more likely to have symptoms of burnout, fatigue, and recent suicidal ideation [19].

The economic, personal, and work-related factors can be responsible for the burnout of the working staff, resulting in the premature departure from their jobs. This influences the high turnover of costs for administration in replacing trained health workers, lost quality, diminished productivity, and lowered morale [20]. The contributors include workload stress, including working in the proximity of terminal illness, trauma, and deaths. This can sometimes lead to psychological stress ending in friction at work, such as verbal violence and others. These factors can be addressed by gaining insight into factors such as recognition being the critical motivator at work, stability, flexible work schedules, professional growth, and adequate supervision, which play a vital role in the longevity of the working staff. This can eventually lead to enhanced job commitment by the working staff [20].

## 5. Consequences for Physician Health

Physician burnout can lead to severe personal and professional consequences if left unaddressed. We can classify the personal consequences of physician burnout in two categories: physical and psychological. These consequences manifest as symptoms and can range from mild to severe on a case-by-case basis. More broadly, burnout is associated with impairment to physicians, who can complain of feeling tired, exhausted, fatigued, inattentive, and irritable [21,22]. Burnout can also put a physician at increased risk of motor vehicle accidents and near-miss events, even after adjusting for fatigue [23]. Psychologically, physician burnout might contribute to increased incidence of stress, disruptive behavior, mood disorders, and a noted correlation with depression [12,24,25]. The presence of any of these conditions can severely impact a physician’s well-being, disrupting their personal life and decreasing professional efficiency. This creates a slippery slope because it increases the odds of substance abuse with increased alcohol abuse/dependence, especially in surgeons [26]. Most physicians do not acknowledge their symptoms or admit that they can be affected by burnout and refuse to seek, help leading to a two-fold increased risk of suicidal ideation [27,28]. Physicians are at an increased risk of suicide (28–40 per 100,000) compared to the general population (12.3 per 100,000), especially among specialties which make up the “front-line of care” like emergency, primary, and preventive medicine [22]. Although physicians of both genders with burnout have an equally high suicide rate, it is thought-provoking to note the completed suicide rate in female physicians exceeds that of the general population by 2.5 to 4 times. They attempt suicide far less often than women in the general population, but when they do, it is usually a completed suicide rather than an attempted one (American Psychiatric Association (APA) 2018. Abstract 1–227, presented 5 May 2018). Therefore, the suicide rate among female physicians is 2.27 times higher compared to females in the general population. In male physicians, it is 1.41 times higher compared to males in the general population [29,30].

Perhaps the professional consequences of physician burnout can contribute to the failure of interpersonal relationships, increased medical errors, increased risk of malpractice, reduced patient satisfaction, and the quality of care and patient outcomes [22,31]. A physician suffering from burnout is less productive and may even quit at some point, impacting the health care system economically by increasing costs. Among physicians, the degree of perceived control over stressors at work can have both behavioral as well as physiological effects, and this is one of the single most potent predictors of burnout. It is conceivable that reduced sense of control over the work environment has been associated with anxiety, reduced motivation and persistence, depression, longer time needed to solve problems, and the tendency to give up easily [32]. The consequences of physician burnout on patient care, physician health, and the health care system are summarized in Table 1.

## 6. Discussion and Conclusions

Hospital management should take initiatives toward understanding the challenges to overcome clinician well-being, proposing more evidence-based solutions and checking their effectiveness in a timely manner. This includes timely surveys using various established scales that promptly measure the physical, emotional, and mental exhaustion of the clinicians. The data can help in understanding the areas required for improvement and act as a baseline to compare future similar surveys. The human resource management (HRM) should decrease the administrative burden on the physicians, including vitals checking, history taking, prescribing medications, and scheduling appointments. These jobs can be distributed to other medical professionals, like medical assistants, who can be trained to manage these responsibilities. In this manner, physicians can spend more quality time with the patients who would benefit from a therapeutic patient–physician relationship. In intense settings such as critical care or ICU, the managers have to be mindful of the working schedules of the physicians. The physicians are continuously surrounded by morbidity and mortality situations, so the chances of exhaustion are high. Timely-breaks and days off should be carefully planned. Additionally, physicians should be encouraged to engage in relaxation techniques like yoga, meditation, and other relaxation techniques, and also to approach therapists to vent their frustrations. This might help them to return to work with a more even mindset and increase work output effectively.

There are some limitations to our review article that should be taken into consideration. First, the papers included in our review do not cover all the research studies conducted in the past. Thus, our evaluation of physician burnout might not be comprehensive. Secondly, we did not define how physician burnout varies among physicians from different specialties of medicine. Physicians from different specialties might experience and cope with this syndrome differently. Thus, it is important to take this limitation into view. Although this review does present some limitations, it explores the universal phenomenon of physician burnout that is impacting our patient care experience in detail, and the facts presented in this review should be considered in evaluating and solving this global issue.

Physician burnout is a significant problem in the medical profession, and work overload is the main contributor to it. It is common, but reversible and preventable. Burnout has adverse outcomes on physician well-being, patient care, and the health care system. Doctors who keep working despite experiencing signs of burnout are more likely to have decreased work productivity, exhaustion, and poor quality of care when compared to their earlier careers. Additionally, it can also increase the economic burden of training and recruiting new staff members when efficient physicians quit due to inability to handle stress. Hence, there is a need for management to keep a check on doctors’ physical and mental well-being. Self-awareness among physicians can enhance the ability to recognize their vulnerability to burnout, and immediate measures should be taken to overcome and manage fatigue, stress, and accentuate resiliency. This can be addressed by the systematic application of evidence-based interventions, including but not limited to group interventions, mindfulness training, assertiveness training, facilitated discussion groups, and promoting a healthy work environment.

## Figures and Tables

**Table 1 behavsci-08-00098-t001:** Consequences of physician burnout.

Consequences to	Study	Brief Description
Patient care	[12,24]	Professional consequences of physician burnout can contribute to the failure of interpersonal relationships, increased medical errors, increased risk of malpractice, reduced patient satisfaction, and affects the quality of care and patient outcomes.
Health care cost	[8,20]	Physician burnout might result in untimely departure of physicians from their jobs, directly affecting the economy of the administration. They spend more money on recruiting new doctors and training them. In this gap, the quality and amount of the work output is jeopardized.
Physician health	[21,22]	Physicians affected by burnout may complain of feeling tired, exhausted, fatigued, inattentive, and irritable.
	[7]	Burnout also puts a physician at an increased risk of motor vehicle accidents and near-miss events, even after adjusting for fatigue.
	[16,24,25]	Psychologically, physician burnout can contribute to increased incidence of stress, disruptive behavior, mood disorders, and a noted correlation with depression.
	[26]	Increased odds of substance abuse in physicians, with increased alcohol abuse/dependence, especially in surgeons.
	[27,28]	Suicidality and two-fold increase in the risk of suicidal ideation was suggested.
	[29,30]	Suicide rate in female physicians is 2.27 times higher than that of general population.Suicide rate in male physicians is 1.41 times higher than that of general population.

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
