# Peer review of "Factors Related to Physician Burnout and Its Consequences: A Review"

_behavsci, 2018, doi:10.3390/bs8110098_

Round 1
Reviewer 1 Report
The work has improved compared to its initial version.
Simply indicate, that the inclusion of some of the data that this monograph contributes could be considered
https://www.mdpi.com/journal/ijerph/special_issues/workplace_health
Author Response
Thank you for reviewing the study and making it stronger. Our team appreciates this effort.
We did review this special issue before submission, but we could not find as specific to physicians. We found one study in MDPI which is cited #20 https://www.mdpi.com/1660-4601/15/9/1850/htm
Reviewer 2 Report
The paper has been improved a lot.
I would suggest the authors uniform the use of the burnout term (in some instances you say burn-out, while in other say burnout).
I will suggest the authors avoid causal language, due to the fact that most of the studies that they reviewed are not experimental or longitudinal, hence these studies did not prove any causality.
Information about the literature search would be expanded because the authors must provide detailed information about the studies retrieved.
Author Response
Thank you for reviewing the study and making it stronger. Our team appreciates this effort.
I would suggest the authors uniform the use of the burnout term (in some instances you say burn-out, while in other say burnout) - Made the tracked changes
I will suggest the authors avoid causal language, due to the fact that most of the studies that they reviewed are not experimental or longitudinal, hence these studies did not prove any causality. - We did mention such studies and you can refer to our reference list. Yet, we changed the title of the article, considering your recommendation. In the text, all the places are mentioned "contributors" not causation.
Information about the literature search would be expanded because the authors must provide detailed information about the studies retrieved. - Made the tracked changes in Methods